# Effect of NIR Laser Therapy by MLS-MiS Source on Fibroblast Activation by Inflammatory Cytokines in Relation to Wound Healing

**DOI:** 10.3390/biomedicines9030307

**Published:** 2021-03-16

**Authors:** Shirley Genah, Francesca Cialdai, Valerio Ciccone, Elettra Sereni, Lucia Morbidelli, Monica Monici

**Affiliations:** 1Department of Life Sciences, University of Siena, I-53100 Siena, Italy; shirley.genah@student.unisi.it (S.G.); ciccone3@student.unisi.it (V.C.); lucia.morbidelli@unisi.it (L.M.); 2ASA Campus Joint Laboratory, ASA Research Division & Department of Experimental and Clinical Biomedical Sciences “Mario Serio”, University of Florence, I-50139 Florence, Italy; francesca.cialdai@unifi.it (F.C.); elettra.sereni@unifi.it (E.S.)

**Keywords:** wound healing, NIR laser radiation, inflammation, fibroblasts, photobiomodulation

## Abstract

The fine control of inflammation following injury avoids fibrotic scars or impaired wounds. Due to side effects by anti-inflammatory drugs, the research is continuously active to define alternative therapies. Among them, physical countermeasures such as photobiomodulation therapy (PBMT) are considered effective and safe. To study the cellular and molecular events associated with the anti-inflammatory activity of PBMT by a dual-wavelength NIR laser source, human dermal fibroblasts were exposed to a mix of inflammatory cytokines (IL-1β and TNF-α) followed by laser treatment once a day for three days. Inducible inflammatory key enzymatic pathways, as iNOS and COX-2/mPGES-1/PGE2, were upregulated by the cytokine mix while PBMT reverted their levels and activities. The same behavior was observed with the proangiogenic factor vascular endothelial growth factor (VEGF), involved in neovascularization of granulation tissue. From a molecular point of view, PBMT retained NF-kB cytoplasmatic localization. According to a change in cell morphology, differences in expression and distribution of fundamental cytoskeletal proteins were observed following treatments. Tubulin, F-actin, and α-SMA changed their organization upon cytokine stimulation, while PBMT reestablished the basal localization. Cytoskeletal rearrangements occurring after inflammatory stimuli were correlated with reorganization of membrane α5β1 and fibronectin network as well as with their upregulation, while PBMT induced significant downregulation. Similar changes were observed for collagen I and the gelatinolytic enzyme MMP-1. In conclusion, the present study demonstrates that the proposed NIR laser therapy is effective in controlling fibroblast activation induced by IL-1β and TNF-α, likely responsible for a deleterious effect of persistent inflammation.

## 1. Introduction

Any injury or infection triggers an inflammatory reaction via cytokines deriving from platelet degranulation and pathogen-associated molecular patterns. Moreover, in both cases, damaged cells release reactive oxygen species and non-specific factors which contribute to activate the inflammatory response in cells of the innate immune system, fibroblasts, and epithelial and endothelial cells [1]. The induction of the inflammatory response triggers a cascade of events mediated by recruitment, proliferation, and activation of several cell populations, primarily immune and stromal cells, as well as further release of cytokines, vasoactive factors, and growth factors that all together contribute to the repair process [2,3].

Therefore, the correct progression of any acutely occurring inflammatory reaction is a key factor in the path leading to successful healing, which consists in repair/regeneration of damaged tissues and function recovery. However, the occurrence of alterations in the finely-tuned regulation of inflammation can cause pathologic conditions ranging from healing delay (e.g., chronic ulcers) to fibrosis. Moreover, conditions of tissue stress or altered function can induce an adaptive response, known as parainflammation or low grade chronic inflammation, which is an intermediate condition between basal homeostasis and acute inflammation, and is associated with serious diseases, including obesity, diabetes, atherosclerosis, asthma, and neurodegenerative diseases [4].

Inflammation is regulated by a plethora of cell populations, biochemical, and physical factors, but it is widely recognized that the cross-talk between macrophages and fibroblasts, and their ability to assume different phenotypes play a crucial role in determining not only the evolution of inflammation, but also the subsequent stages of the healing process. 

During inflammation, macrophages shift from a pro-inflammatory phenotype (the so called M1), characterized by massive production of pro-inflammatory molecules, to an anti-inflammatory phenotype (the so called M2), which secretes suppressors of cytokine signaling, passing through intermediate phenotypes [3,5,6,7].

In response to pro-inflammatory mediators, resident fibroblasts or circulating fibrocytes become the protagonists of the stromal activation and transdifferentiate in myofibroblasts, their activated counterpart. Many pro-inflammatory mediators are implicated in fibroblast activation, migration, proliferation, and transdifferentiation, including the cytokines tumor necrosis factor-α (TNF-α), interleukin-1 (IL-1), interleukin-6 (IL-6), and the growth factors platelet derived growth factor (PDGF) and fibroblast growth factors (FGFs). Activated fibroblasts and other mesenchymal cells engage a crosstalk, which also reinforces the local immune response due to the induction of vasodilation throught production of nitric oxide (NO) and prostanoids, and stimulates angiogenesis via vascular endothelial growth factor (VEGF) production [8,9]. 

In a normal evolution of the process, the turning off of the inflammatory response, mediated by the shift of the macrophage phenotype from M1 to M2, opens the way to the remodeling phase, which is dominated by fibroblasts through the production of extracellular matrix (ECM) proteins and matrix metalloproteinases (MMPs) [10]. A well-timed resolution of inflammation is crucial for successful restoration of tissue architecture and function, while persistence of macrophage-fibroblast activation state, with excessive production of pro-inflammatory agents by fibroblasts and further recruitment of immune cells, leads to altered repair processes, from chronic wounds to fibrosis and scarring [11,12].

In summary, activated macrophages induce the stimulation of fibroblasts via production of transforming growth factor-β (TGF-β), TNF-α, IL-1, and other cytokines. In turn, activated fibroblasts can modulate the recruitment and behavior of immune cells via release of cytokines and vasoactive factors as NO and prostanoids. Activated fibroblasts, or myofibroblasts, regulate tissue remodeling by combining their ability to synthesize ECM proteins and that of assuming contractile properties [13,14]. Contractile activity of myofibroblasts increases ECM stiffness. In turn, ECM stiffness is, together with TGF-β1, among the most important factors in inducing myofibroblast differentiation and persistence. Therefore, inflammation dysregulation can generate a feed-forward loop with detrimental effects [15]. Therefore, the control of inflammation and fibroblast activation is crucial to obtain satisfactory morpho-functional recovery and avoid defective healing.

Whatever the cause of inflammation (wound, trauma, infection), at the tissue level it is characterized by redness, heat, oedema, pain, and loss of function.

In current clinical practice, a series of steroidal and nonsteroidal anti-inflammatory drugs can be used to control inflammation and the associated oedema and pain [16]. However, side effects, or even opposite effects on wound healing and other conditions inducing inflammation, limit their use, especially considering long-term therapy, raising the need for alternative countermeasures [16]. Moreover, anti-inflammatory strategies focused on a specific target (e.g., TNF-α) did not produce the desired results [17]. Several physical therapies and devices aimed to favor the healing process through the control of inflammation and fibroblast behavior have been proposed, such as ultrasound, laser therapy, electrical stimulation, and vacuum-assisted closure [16,18,19]. Studies aimed at elucidating the effectiveness of these therapies in controlling inflammation and the deriving fibroblast activation might strengthen their use.

Laser therapy, currently called photobiomodulation therapy (PBMT), is one of the most widely applied to manage many different diseases characterized by acute or chronic inflammation. The benefits of PBMT in terms of anti-inflammatory [20], anti-pain [21,22,23], and anti-oedema [24] properties are widely documented in literature. Moreover, PBMT enhances cell energy metabolism and promotes anabolic and repair processes [25]. Further, PBMT has been shown to stimulate angiogenesis and collagen remodeling [16,26]. 

PBMT, being safe, non-invasive, and non-time-consuming (short-duration application) is also well accepted by patients. 

Over the last years, new molecular insights into the action mechanisms of PBMT have been obtained. In particular, it has been demonstrated that in chronic inflammatory conditions, such as those related to periodontal diseases and osteoarthritis, PBMT is effective in reducing the expression of pro-inflammatory genes (TNF-α, IL-1β, IL-6, IL-8) through the downregulation of NF-ĸB signaling pathway via cAMP increase [27].

Another PBMT effect, which can be relevant in the evolution and outcome of the inflammatory response, is to induce a decrease in matrix metalloproteinases (MMPs) expression, as it has been recently demonstrated in an in vitro model of osteoarthritis [28]. MMPs are an important family of proteinases, able to degrade extracellular matrix components and covering a broad range of tasks in inflammation, acquired immunity, defense from injury and repair. MMPs are always present in acute and chronic, physiological and pathological inflammatory processes, and experimental evidence suggests that they can protect against or contribute to pathological evolution of inflammation [29,30].

Despite the abundant literature on the ability of PBMT to control inflammation, promote healing mechanisms, and counteract scarring, the effects that laser emissions commonly used in PBMT exert on fibroblasts activated by a strong and persistent inflammatory stimulation have not been clearly defined. In fact, for the most part, studies used in vitro models of fibroblasts in the basal state.

Therefore, the present study was aimed at investigating the effect of PBMT on activated fibroblasts. An in vitro model of fibroblasts, activated by exposure to inflammatory stimuli, was characterized for morphological features, canonical inflammatory and vasoactive cascades (inducible NO synthase and cyclooxygenase (COX)/prostaglandin synthase enzymes), and outcomes on angiogenesis and ECM remodeling. Then, the effectiveness and underlying molecular mechanisms of a high power, dual wavelength NIR laser source in reducing fibroblast inflammatory phenotype was investigated.

## 2. Materials and Methods

### 2.1. Cell Cultures

Normal human dermal fibroblasts (NHDF) were purchased from Lonza (Verviers, Belgium) and grown in Fibroblast Growth Basal Medium (FBS; Lonza, Basel, Switzerland) containing 10% Fetal Bovine Serum (FBS; Hyclone, Euroclone, Milan, Italy), 2 mM glutamine, 100 units/mL penicillin, and 0.1 mg/mL streptomycin (Merck KGaA, Darmstadt, Germany). Cells were cultured at 37 °C with 5% CO_2_ in Petri dishes and were split 1:3 twice a week until passage 10. 

### 2.2. In Vitro Model of Inflammation

Cells (1 × 10^4^) were seeded in 24-multiwell plates and allowed to adhere (when immunofluorescence analysis were planned, cells were seeded on 13 mm diameter glass coverslips placed inside the 24-multiwell plates). After 24 h, complete culture medium was replaced by fresh complete culture medium supplemented with a mix of IL-1β (10 ng/mL; #201-LB/CF R&D System, Minneapolis, MN, USA) and TNF-α (10 ng/mL; #201-LB/CF and #410MT, R&D System, Minneapolis, MN, USA). Cells were maintained with the cytokines mix for 48 h. Control samples were treated in the same way, omitting the cytokines mix. 

### 2.3. Laser Treatment

At the end of the 48 h of stimulation with cytokines mix, the medium of all samples was replaced by a fresh complete culture medium. Then, samples which had been previously stimulated with the cytokines mix were divided into two groups: A “treated group”, that received laser irradiation and an “untreated group” that was not laser irradiated. Laser treatment was performed with a Multiwave Locked System laser (MLS-MiS, ASA S.r.l., Vicenza, Italy) widely used in clinics. It is a class IV, NIR laser with two synchronized sources (laser diodes): The first one is a pulsed laser diode emitting at 905 nm wavelength, with peak power from 140 W ± 20% to 1 kW ± 20% and pulse frequency varying in the range 1–2000 Hz; the second laser diode emits at an 808 nm wavelength and can operate in continuous (max power 6 W ± 20%) or frequent (repetition rate 1–2000 Hz, 50% duty cycle) mode. The two laser beams work simultaneously and synchronously, and the propagation axes are coincident. 

For laser exposure, only 6 wells of 24-well plates contained cells. The wells surrounding those with cells were filled with black cardboard to avoid light diffusion and reflection. The exposure was performed by placing the plate inside a holder, which allows the positioning of the laser handpiece at a 1.5 cm distance from the bottom of the wells, so that the spot of the two laser beams, impinging perpendicular to the sample surface, had the same diameter as a well (13 mm). Cells were irradiated for 10 sec with the following parameters: 10 Hz repetition rate; 50% int (mean power 1840 mW); peak power 1 kW ± 20%, fluence 5.19 J/cm^2^. All treatments were performed under laminar flow hood at room temperature. The samples belonging to the untreated group were prepared and kept under the same conditions used for the exposed samples, except for laser irradiation. 

### 2.4. Experiment Design

The following samples were prepared, analyzed, and compared: (i)samples stimulated with a mix of IL-1β and TNF-α for 48 h and then exposed to 3 laser treatments, repeated once a day, for 3 consecutive days under sterile conditions (CYKs + LASER Group);(ii)samples stimulated with a mix of IL-1β and TNF-α for 48 h and not exposed to laser treatments CYKs Group);(iii)samples not stimulated with a mix of IL-1β and TNF-α for 48 h and not exposed to laser treatments (CTRL Group).

For immunofluorescence analysis, an additional experimental group was included, namely cells exposed to laser treatment alone (LASER Group). 

Six hours after the third laser treatment, all samples were prepared for the subsequent analysis described in the following paragraphs.

### 2.5. Immunofluorescence Analysis

Cells grown on glass coverslips and treated as previously described, were fixed for 5 min with ice cold acetone. Unspecific binding sites were blocked with PBS containing 3% bovine serum albumin (BSA; Sigma-Aldrich, St. Louis, MO, USA) for 1 h at room temperature. Then, cells were incubated overnight at 4 °C with specific anti-NF-kB (1:50; #sc-372, Santa Cruz, Dallas, TX, USA), anti-cyclooxygenase-2 (COX-2) (1:100; #TA313292, Origene, Rockville, MD, USA), anti-VEGF (1:50; #sc-57496, Santa Cruz), anti-α actin (1:100; #MAB1501X, Millipore, Billerica, MA, USA), anti α-smooth muscle actin (α-SMA) (1:100; #CBL171, Chemicon^®^ by Thermo Fisher Scientific, Waltham, MA, USA), anti-tubulin (1:100; #05-829, Millipore, Billerica, MA, USA), anti-collagen I (1:100; #MAB3391, Millipore, Billerica, MA, USA), anti-fibronectin (FN) (1:100; #MAB1926-I, Millipore, Billerica, MA, USA), anti-MMP-1 (1:100; #MAB13439, Millipore, Billerica, MA, USA), and anti-α5β1 integrin (1:100; #MAB1999, Millipore, Billerica, MA, USA) primary antibodies properly diluted in PBS with 0.5% BSA. After washing three times with PBS-0.5% BSA, samples were then incubated for 1 h at 4 °C in the dark with: Alexa Fluor 555™ conjugated secondary antibodies [specifically: Anti-mouse IgG (#A-21422, Invitrogen™ by Thermo Fisher Scientific) for anti-NF-kB and anti-VEGF antibodies and anti-rabbit IgG (#A-21428, Invitrogen™ by Thermo Fisher Scientific) for anti-COX-2 antibody] and fluorescein isothiocyanate (FITC) conjugated specific secondary antibody [specifically: Anti-mouse IgG (#AP124F, Millipore) for anti α-SMA, anti-tubulin, anti-collagen I, anti-fibronectin, anti-MMP-1, anti-α5β1 integrin primary antibodies]. All secondary antibodies were diluted 1:200 in PBS with 0.5% BSA. Cells incubated with anti-α actin antibody did not need incubation with the secondary antibody since a mouse anti-actin Alexa Fluor^®^ 488 conjugated was used. Again, samples were washed three times and then mounted on glass slides using Fluoromount™ aqueous mounting medium (Sigma-Aldrich St. Louis, MO, USA) [31]. In samples of incubated anti-NF-kB, anti-COX-2, and anti-VEGF, before mounting, nuclei were marked with DAPI (#D9542, Sigma-Aldrich, St. Louis, MO, USA) diluted 1:5000 in PBS with 0.5% BSA for 30 min at room temperature. The fluorescent signal of samples stained with anti-NF-kB, anti-COX-2, and anti-VEGF antibodies was acquired using a Leica TCS SP5 laser scanning confocal microscope (Leica, Wetzlar, Germany). All other samples were evaluated by an epifluorescence microscope (Nikon, Florence, Italy) at 100x magnification and imaged by a HiRes IV digital CCD camera (DTA, Pisa, Italy). Based on the CCD images, a relative immunofluorescence quantification was carried out by image analysis routines (ImageJ 1.53 analysis software, National Institutes of Health, Bethesda, MD, USA) for samples stained with anti-collagen I and anti-α5β1 integrin antibodies. After appropriate thresholding to eliminate background signal and creation of a proper image mask, a pixel intensity histogram was acquired. 

### 2.6. Western Blot

Cells derived from the different experimental conditions, were detached from 24 multi-well plates, collected in 15 mL tubes, and lysed with CelLytic^TM^ MT Cell Lysis Reagent supplemented with 2 mM Na_3_VO_4_ and 1X Protease inhibitor cocktail for mammalian cells (Sigma-Aldrich). Cell lysates were centrifuged at 16000× *g* for 20 min at 4 °C, and the supernatants were then collected. Protein concentration was determined using the Bradford protein assay (Sigma-Aldrich). Electrophoresis with equal amounts of proteins (50 μg) was carried out in NuPAGE^TM^ 4–12% Bis-Tris precast Gels (Thermo Fisher Scientific) as previously reported [32]. 

Proteins were transferred onto nitrocellulose membranes, blocked for 1 h in a PBS–0.05% Tween solution (Sigma-Aldrich) supplemented with 5% (wt/vol) of Blotting-Grade Blocker (Bio-Rad, Hercules, CA, USA). Membranes were then incubated overnight at 4 °C with the primary antibodies properly diluted in PBS–0.05% Tween solution supplemented with 1% (wt/vol) of Blotting-Grade Blocker: anti-inducible NO synthase (iNOS) (1:250; #sc-7271, Santa Cruz), anti-COX-2 (1:1000; #160106, Cayman Chemical, Ann Arbor, MI, USA), and anti-microsomal prostaglandin E synthase-1 (mPGES-1) (1:500; #160140, Cayman Chemical). Immunoblots were washed three times with PBS–0.05% Tween solution and then incubated for 1 h with the respective species-specific secondary antibody conjugated with horseradish peroxidase HRP (Promega, Madison, Wisconsin, US) diluted 1:2500 in PBS–0.05% Tween solution. The membranes were finally incubated with SuperSignal^TM^ West Pico PLUS chemiluminescent Substrate (Thermo Fisher Scientific), and the immunoreaction was revealed by ImageQuant LAS 4000 chemiluminescence system (GE Healthcare, Chicago, IL, USA). Results were normalized to those obtained by using an antibody against β-Actin (#A5441, Sigma-Aldrich) diluted 1:10,000 in PBS–0.05% Tween solution. 

Immunoblots were analyzed by densitometry using Image J software, and the results, expressed as arbitrary density units (A.D.U.), were normalized to β-Actin. 

### 2.7. Immunoassays for Prostaglandin E-2 and VEGF Quantification

Conditioned media were collected at the end of the experiment, frozen, and stored at −80 °C until use. Prostaglandin E-2 (PGE-2) and VEGF levels were measured using ELISA kit: Prostaglandin E_2_ ELISA kit-Monoclonal (Cayman Chemical, Ann Arbor, Michigan, US) and VEGF ELISA kit (R&D Systems, Minneapolis, MN, USA), respectively, following the manufacturer’s instructions. Dosing of each sample was performed in double, and PGE-2 and VEGF levels were expressed as (pg/mL). 

### 2.8. Statistics

Three different experiments were carried out in triplicate. Data are reported as means ± SD. Statistical significance was determined using two-sided Student’s *t* test. A *p* value lower than 0.05 was considered statistically significant. For immunofluorescence analysis, at least 30 cells per slide were scored in 10 random fields/slide.

## 3. Results

### 3.1. Set up of an “In Vitro” Inflammatory Model in Fibroblasts Cultures

The human dermal fibroblasts NHDF have been treated with a mix of cytokines (IL-1β and TNF-α, each at 10 ng/mL) for 24 h and 48 h, then the occurrence of inflammatory features depending on the exposure time has been evaluated. 

Microsomal PGE synthase-1 (mPGES-1), the pivotal inducible enzyme of the prostanoid inflammatory pathway, was evaluated by western blot after 24 h and 48 h of stimulation. A consistent rise in mPGES-1 was observed after both 24 h and 48 h, the increase being more evident at a longer time of exposure (Figure 1, upper panel, 0.8 ± 0.2 and 4.8 ± 0.9 fold increase of mPGES-1 expression in the presence of cytokines with respect to control, at 24 h and 48 h, respectively). The up-regulation of the mPGES-1 enzyme generated a significant increase in the final product, prostaglandin E2 (PGE-2), released by fibroblasts in the conditioned medium, documenting an activation of the enzymatic cascade (Figure 1, lower panel). Based on these results, the stimulation time of 48 h was chosen for further experiments.

### 3.2. Effect of Laser Treatment on Inflammatory Phenotype in Fibroblasts

#### 3.2.1. Expression of Inflammatory Markers

In order to evaluate whether laser treatment could affect the inflammatory model described above, samples, after the stimulation with cytokine mix, were exposed to laser radiation according the following experimental protocol: NHDFs were treated with cytokine mix (IL-1β and TNF-α, each at 10 ng/mL) for 48 h; then, culture medium was replaced by fresh culture medium, and samples were divided into the following groups: (i) CYKs + LASER Group—samples previously stimulated with the cytokine mix and then exposed to laser treatment (3 treatments, repeated once a day, for 3 consecutive days); (ii) CYKs Group—samples previously stimulated with the cytokine mix and not exposed to laser treatment; (iii) CTRL Group-samples not stimulated with the cytokine mix and not exposed to laser treatment. Six hours after the third laser treatment (T = 126 h), all the samples were recovered and iNOS, COX-2 and mPGES-1 protein expression was evaluated by western blotting.

Following the exposure to inflammatory cytokines, a significant up-regulation of the inflammatory enzymes was observed (Figure 2). The group stimulated with cytokines and then treated with lasers showed a strong decrease in inflammatory enzyme expression compared to the group stimulated only with cytokines. For iNOS and COX-2, the decrease reached statistical significance (Figure 2).

To validate data obtained by Western blot, the main product of prostanoid enzymatic cascade, PGE-2, was measured in NHDF conditioned media recovered from the samples 6 h after the third laser treatment (T = 126 h). The cytokine mix-stimulated fibroblasts showed a significant increase in PGE-2 released in the medium in comparison with control samples (209 pg/mL in basal condition and 11000 pg/mL after 48 h of stimulation with the inflammatory mix). Although the resulting data were not significant, laser exposure reduced PGE-2 levels with a clear trend towards damping of the prostanoid pathway (Figure 3, upper panel). 

Additionally, conditioned media were assessed for the release of the angiogenic factor VEGF, involved in neovascularization and granulation tissue formation. While inflammatory stimuli significantly increased VEGF levels, laser exposure strongly reduced VEGF availability in the medium, being the levels well below the basal, unstimulated condition (Figure 3, lower panel).

Ultimately, the modulation of the inflammatory response at the cellular level was evaluated through confocal microscopy. Inflammation is a protective response characterized by a series of reactions, such as vasodilation and recruitment of immune cells to the site of injury. NF-κB is an inducible transcription factor, responsible for the activation of genes involved in this process, including COX-2 and VEGF [33]. The localization of the nuclear transcription factor NF-κB and the intensity of the fluorescent signal given by the expression of its downstream genes COX-2 and VEGF were analyzed in the samples described above. In control samples, the transcription factor seemed to remain outside the nucleus, since the fluorescent signal was mainly cytoplasmic (Figure 4a). NHDF stimulation with IL-1β and TNF-α induced a consistent increase in the expression of NF-κB, as evidenced by a higher fluorescence intensity. Furthermore, in many cells, a change in the localization was observed, with accumulation of the signal at the nuclear level (Figure 4b; white arrows). In samples stimulated with the cytokine mix and then treated with laser, a clear decrease in intensity of the signal linked to the transcription factor was observed, although some cells with NF-κB located in the nucleus (Figure 4c; white arrows) were still present. In cells treated with laser alone, the presence of some NF-κB punctuation at nuclear level was observed (Figure 4d).

A similar trend was described for COX-2. The enzyme expression resulted strongly enhanced by the cytokine mix (Figure 4f) in comparison with unstimulated and laser alone controls (Figure 4e,h), where the fluorescence signal was weak and located in the cytoplasm. In samples stimulated with the cytokine mix and then treated with laser, the signal was similar to that observed in control (Figure 4g). However, in the last condition, mixed cell populations were noticed, some still over-expressing the enzyme and others returned to control levels. Similarly, the VEGF signal also followed a modulation superimposable to that of COX-2, presumably dictated by the transcription factor NF-kB. VEGF labelling, not affected by laser alone (Figure 4k), increased by stimulation with cytokines mix (Figure 4j). Following laser treatment, the cytokine-induced VEGF intensity completely returned to the basal levels (Figure 4l), confirming the data obtained by protein dosage carried out on conditioned media, as previously illustrated (Figure 3, lower panel).

#### 3.2.2. Morphology and Cytoskeleton Organization

The stimulation with the mix of IL-1β and TNF-α produced a marked change in cell morphology. The organization of the microtubules, which control cell architecture, changed as well. In the control samples, fibroblasts were generally star-shaped and spread on the substrate. The specific labeling for tubulin showed the well-known radial distribution of microtubules which branch off from a nucleation center (Figure 5a), usually anchored at the centrosome and the Golgi apparatus [34]. In the stimulated samples, the cells appeared spindle-shaped, elongated, with a dense, longitudinal microtubule network [34], where it was difficult to distinguish a nucleation center (Figure 5b). In the samples first stimulated and then treated with laser, fibroblasts regained a shape similar to the controls, with a clearly distinguishable microtubule nucleation center and radially organized microtubules (Figure 5c).

As regards actin distribution and organization, control fibroblasts showed a perinuclear area rich of G-actin, a network of very thin microfilaments distributed in the cell cytoplasm, and a thin actin layer placed close to the plasma membrane (Figure 5d). In the stimulated fibroblasts, F-actin was predominant, with microfilaments arranged in parallel and thicker in comparison with those observed in control cells (Figure 5e). As already noted for tubulin, also in the case of actin, the stimulated cell samples which were then exposed to laser radiation recovered a condition similar to the control cells, with G-actin thickened in the perinuclear area, few very thin microfilaments and a thin actin layer close to the cell membrane (Figure 5f). 

Alpha-smooth muscle actin (α-SMA) is the actin isoform that predominates within smooth-muscle cells. Its expression generally increases in the transition fibroblast-myofibroblast. In fact, myofibroblasts acquire a contractile phenotype, which is responsible for merging the wound edges in the healing process. Therefore, α-SMA is considered a marker of myofibroblast differentiation. In control samples, α-SMA staining revealed some stress fibers, which completely disappeared in fibroblasts stimulated with IL-1β and TNF-α, where the fluorescence signal coincided with the nucleus and was detectable only in the nuclear area (Figure 5g,h). The samples exposed to laser radiation after the cytokine mix stimulation showed an intermediate situation. The signal in the nuclear area was still detectable, but fibers organized in parallel appeared (Figure 5i). 

#### 3.2.3. Extracellular Matrix Proteins and Membrane Integrin

Integrins are cell surface receptors which control various cellular functions. Integrin receptors connect the cell cytoskeleton with the ECM proteins, thus being involved in signaling changes of the extracellular microenvironment and leading to cellular responses. In particular, α5β1 integrin is a fibronectin receptor and has a well-defined role in cell adhesion, migration, and matrix formation, which are functions of crucial importance in physiological and pathological processes such as wound healing and fibrosis. In the control samples, α5β1 clusters were located at focal adhesion points mostly in the perinuclear area, along cellular protrusions, and at their ends, generally arranged parallel to the major axis of the cells (Figure 6a). In the samples stimulated with the cytokine mix, the expression of the integrin significantly increased (Figure 6b). In these samples, half of the cells still retained a morphology similar to the control (star-shaped and spread), but showed a higher density of integrin clusters with centripetal distribution in the perinuclear area. In the other half of the cells, characterized by spindle-shaped and elongated morphology, the α5β1 clusters became point-like, smaller, distributed in the perinuclear area, and at lateral intercellular surfaces forming cell–cell contact points (Figure 6b). After laser treatment, fibroblasts appeared similar to the controls, both for the signal intensity and distribution of α5β1 clusters (Figure 6c).

Through its interaction with different cell types, cytokines, and other ECM molecules, and facilitating collagen fibrogenesis by scaffolding action, fibronectin plays a preeminent role in both wound healing and scarring [35,36]. Similarly to its receptor α5β1, fibronectin significantly increased in fibroblast cultures stimulated with IL-1β and TNF-α, when compared to unstimulated controls, and formed a dense extracellular network of fibrils (Figure 7a,b). After laser treatment, fibronectin expression returned to the basal levels observed in control cells with evident reduction of extracellular fibrils (Figure 7c,d). 

In addition, the synthesis of collagen I, one of the most abundant ECM components, was significantly enhanced by the exposure to the cytokine mix in comparison with control unstimulated cells (Figure 8a,b). Interestingly, stimulated fibroblasts showed an intracellular accumulation of collagen I, apparently in the endoplasmic reticulum and/or Golgi apparatus, while the protein was not released in the extracellular environment (Figure 8b). Cytokine-mix stimulated fibroblasts exposed to the laser treatment revealed a collagen I signal similar to that observed in the control, both for distribution and fluorescence intensity (Figure 8c,d).

Matrix metalloproteinases (MMPs) are endopeptidases that can degrade the ECM proteins. They have important roles in fundamental physiological processes, such as embryonic development, morphogenesis, and tissue remodeling, and are involved in a number of diseases. MMPs are present in both acute and chronic wounds, where they regulate ECM degradation/deposition that is essential for wound healing. The excess protease activity can lead to chronic nonhealing wounds [37].

Specifically, MMP-1 is able to degrade collagen types I, II, and III. Similarly to fibronectin and collagen, also MMP-1 significantly increased in fibroblasts stimulated with the cytokine mix (Figure 9b), compared to non-stimulated controls (Figure 9a). Laser treatment counteracted the effect of the cytokine mix and reported MMP-1 expression to a level comparable to that found in the basal state (Figure 9c).

## 4. Discussion

The cytokines IL-1β and TNF-α have already been used at different concentrations, individually or in association, to stimulate an inflammatory response in various cell types, dermal fibroblasts included [28,38,39]. In this study, IL-1β and TNF-α have been used jointly to induce a pro-inflammatory phenotype in dermal fibroblasts, with the aim to investigate if PBMT delivered via a dual-wavelength NIR laser system (MLS-MiS) was effective in counteracting cell inflammatory response and modulating fibroblast functions involved in stromal activation, wound healing, and its alterations, which can lead to chronic ulcers or fibrosis. Preliminary experiments performed to define the protocol for preparing the in vitro model of inflammation in dermal fibroblast cultures showed that both 24 h and 48 h exposure to IL-1β and TNF-α produced a significant increase in the inducible enzyme mPGES-1 and in the release of its product PGE-2. The increase in mPGES-1 was higher after 48 h, while the increase in PGE-2 was similar at 24 h and 48 h. Therefore, a stimulation time of 48 h was chosen for the subsequent experiments in which non-stimulated controls, samples stimulated with IL-1β and TNF-α, and samples exposed to laser radiation after the stimulation with the inflammatory cytokines were compared for their morphology, inflammatory profile, and expression of molecules involved in ECM remodeling. In stimulated samples, the inflammatory signals iNOS, COX-2, and mPGES-1 significantly increased in comparison with non-stimulated controls, in agreement with data reported in literature [40] and supporting the validity of the inflammatory model used. Samples stimulated and then treated with PBMT showed a significant decrease in iNOS and COX-2, compared to the stimulated but non-laser irradiated samples. The mPGES-1 level and that of the final product PGE-2 decreased, but not significantly, suggesting a multimodal action of PBMT, which could act at different cellular levels (gene transcription, protein expression, and localization), as demonstrated by the reported results. 

Modulation of the three mediators mentioned above is closely related to their upstream activator NF-κB, an inducible transcription factor which is activated upon binding of pro-inflammatory cytokines, such as TNF-α, to their membrane receptors. In basal conditions, NF-κB is sequestered in the cytoplasm by a family of inhibitory proteins. Following inflammatory stimuli, this protein moves to the nucleus, binds to specific elements on DNA, and recruits cofactors for the transcription of target genes *iNOS, COX-2*, and *mPGES-1* [41]. In the presence of inflammatory stimuli, the co-localization of the transcription factor NF-κB within the nucleus, observed by immunofluorescence, correlates with an increased expression of iNOS, COX-2, and mPGES-1 at the cytoplasmic level and a consequent increase of PGE-2 released in the extracellular medium. Following laser treatment, these values are significantly reduced, demonstrating the effectiveness of the laser source and treatment parameters used in counteracting the inflammatory response.

The anti-inflammatory properties of the NIR source used had already been highlighted by a proteomics study on laser-irradiated myoblasts, in which a marked increase in NLRP10, a strong inhibitor of the inflammasome, and in turn of IL-1β and interleukin-18 (IL-18) release, was observed [25]. These data are in agreement with previous studies showing the effectiveness of red and NIR radiation in reducing the inflammatory signals both in fibroblast cultures and at the wound level [42,43]. 

As previously mentioned, inflammation is a protective response characterized by a series of reactions modulated by the master regulator NF-kB, whose gene targets are involved both in the recruitment of immune cells to the site of injury and in vasodilation [33]. In this study, fibroblast production of VEGF at the cytoplasmic level and its secretion in the extracellular milieu were therefore analyzed. Compared to untreated controls, IL-1β and TNF-α stimulation of fibroblasts induced an increase in VEGF production and release in culture medium, that further confirms the validity of the model used for the present study. The proinflammatory cytokines-induced enhancement in VEGF levels is widely documented in vitro [44], and it occurs in vivo in chronic inflammatory diseases as well as in acute inflammatory response to infections and injuries. VEGF is produced by the most part of cell populations involved in wound healing, as platelets, immune cells (neutrophils and macrophages), fibroblasts, and endothelial cells, and reaches the maximum concentration during the proliferative phase. In the wound, VEGF promotes angiogenesis [45] and influences re-epithelialization and collagen deposition through stimulation of keratinocytes and fibroblasts [46]. However, if in a proper inflammatory response VEGF upregulation is needed to promote angiogenesis, excessive or persistent inflammation and VEGF production can lead to fibrosis and should be controlled. Laser treatment subsequent to IL-1β and TNF-α stimulation abolished the cytokine-mediated VEGF increase and brought VEGF levels back to values even lower than those seen in unstimulated controls. In literature, a modulation of VEGF expression following irradiation with red-NIR wavelengths has been described, the final effects depending on irradiation parameters and experimental models used [42,47,48]. The decrease observed in the present study, irradiating activated fibroblasts with the source and parameters described, further supports the strong anti-inflammatory action of the proposed laser treatment. At the same time, the results on cells exposed to laser alone, which did not affect markers of fibroblast activation, substantiated the safety of laser irradiation on quiescent unstimulated cells.

IL-1β and TNF-α treatment induced also noticeable morphological changes with cytoskeletal rearrangements in the network of microtubules and actin microfilaments.

Microtubules form a scaffold which controls cell shape, intracellular transport, signaling, and organelle positioning. Microtubules are stiff and intrinsically polarized structures built of directionally aligned αβ-tubulin dimers. Their “minus” end is anchored at so called microtubule-organizing centers, whereas the “plus” ends can extend or shrink and interact with different intracellular structures [49]. Cells able to readily reorient their polarity axis, such as fibroblasts, generally present a radially organized microtubule array, whose changes are mutually related to cell polarity and can mechanically contribute to cell asymmetry by promoting cell elongation [34]. The results of the present study show that, following cell activation by cytokines, changes in microtubule density and orientation occurred and probably contributed to the observed cell elongation. 

Following IL-1β and TNF-α stimulation, the actin filament network changed as well. Density and thickness of actin filaments increased, while their distribution underwent a rearrangement, giving rise to an array of filaments aligned parallel to the major cell axis. Considering that microtubules are connected to the layer of actin filaments close to the cell membrane through a complex of adaptor proteins often associated with focal adhesions, the changes in microtubule and actin filament networks are probably interrelated, and the rearrangement in α5β1 membrane integrin distribution observed in cytokine-stimulated cells further support this hypothesis. The inhibition of the inflammatory response due to laser treatment led to a partial recovery of the basal cytoskeleton organization in fibroblasts irradiated after cytokine stimulation. Cytoskeleton changes connected with effects produced by NIR laser irradiation have been previously described in different cell models [25,50] and depend on cell type, cell status, parameters, and sources used. 

α-SMA is one of the six actin isoforms. Together with β- and γ-actin isoforms, α-SMA is expressed in some fibroblast/myofibroblast subpopulations in the basal state, where it has a cytoplasmic localization and participates in stress fiber formation. α-SMA is strongly induced by mechanical stress and TGF-β1 in activated myofibroblasts [51], therefore it is generally considered a marker of fibroblast-myofibroblast transdifferentiation. In the present study, dermal fibroblasts stimulated with IL-1β and TNF-α showed α-SMA expression seemingly concentrated in the nucleus, while cytoplasmic stress fibers, to some extent present in unstimulated cells, completely disappeared in the stimulated ones. These results are consistent with in-depth investigations on α-SMA distribution and roles carried out in the last two decades. It has been demonstrated that the apparent nuclear localization is due to deep invaginations of the nuclear membrane filled of α-SMA [52]. The role of the nuclear invaginations is currently quite completely unknown, but it has been hypothesized that these structures could be involved in cellular and nuclear mechanotransduction, nuclear transport, calcium signaling, cell differentiation [52,53,54]. Moreover, in agreement with our results, it has been found that TNF-α suppresses α-SMA expression and stress fiber formation in dermal fibroblasts and that persistent inflammation, mediated by TNF-α, might prevent normal matrix deposition and myofibroblast-dependent wound contraction mediated by TGF-β1 in physiological wound healing [55]. The inhibition of stress fiber formation would turn the cells into a phenotype more migratory and less able to generate tractional forces [51], with possible consequences and delay in the healing process. 

Additionally, in the case of α-SMA, NIR laser treatment after IL-1β and TNF-α partially prevented the cytokine effect and some stress fibers reappeared inside the cells. α-SMA expression following red- or NIR-laser treatment has been widely studied being connected with the effectiveness of laser therapy in promoting wound healing and avoiding scarring. The results have been controversial, showing both down- and up-regulation of α-SMA expression [56,57,58]. This variability in results is possibly due to the many different models (from cell cultures to animal models both normal and representing serious diseases, such as diabetes), laser sources, treatment protocols and parameters, and times at which analysis of α-SMA expression was performed. Interestingly, some studies in which the analysis of α-SMA expression was performed at different healing times after laser treatment showed that α-SMA expression changed in the different healing phases and resulted significantly different from controls only at specific time points [59]. The only unambiguous result is that laser irradiation is able to modulate α-SMA, but the modulation depends on many factors, among which the healing phase and corresponding cell phenotype (e.g., the phenotype of fibroblasts in the inflammatory phase is different from what they assume in the remodeling phase). This means that further studies are needed to develop treatment protocols suitable for the different patient’s conditions and, in case of wounds, healing phase. However, the data of the present study indisputably demonstrate that, even at the cytoskeletal level, the source and the treatment parameters used are effective in counteracting the changes induced by cytokine stimulation, thus returning the cells to the basal state.

Compared to controls, the IL-1β and TNF-α stimulated fibroblasts showed increased fibronectin (FN) expression and assembly observed in the same samples. The increase in FN, a major ECM component, could be expected since the pro-inflammatory cytokines IL-1β and TNF-α, together with TGF-β, are considered potent fibrogenic initiators [60]. The increase in expression of α5β1 observed in the same samples is consistent with that of FN, considering that α5β1 is a membrane integrin able of binding FN [61].

A number of studies investigated the expression and role of α5β1 and its ligand FN in fibroblasts during inflammation and wound healing. In the healing process, α5β1-mediated fibroblast-FN interaction is crucial: α5β1 is involved in myofibroblast differentiation and granulation tissue formation by promoting FN assembly in a fibrillar structure. In the granulation tissue, a reduced ability to bind FN via integrin α5β1 might allow fibroblasts to migrate in the early FN-rich matrix and invade the wound [62]. On the other hand, some studies demonstrated that α5β1 integrin is able to confer strong cohesivity to 3D cellular aggregates linking adjacent cells together via FN, and that the FN with its dimeric structure is essential for this process [63]. Moreover, it has been suggested that α5β1-FN interaction contributes to clot retraction [63]. Therefore, α5β1 and FN play a crucial role in wound healing, and alterations in their expression can lead to healing impairment and fibrosis. These conditions can affect ECM remodeling by stimulating collagenase production and stimulating/inhibiting collagen/glycosaminoglycan biosynthesis depending on the target cells and experimental conditions.

The effects of the pro-inflammatory cytokines IL-1β and TNF-α on ECM remodeling and their role in fibrosis have been studied for many years with controversial results. In the present study, the cytokine-stimulated dermal fibroblasts showed increased expression of MMP-1 and collagen I, which have key roles in ECM degradation and building, respectively, thus modulating ECM turnover. In agreement with literature, the intracellular distribution of MMP-1 was associated with mitochondria [64] and, probably, the cytoskeleton. In fact, a relation between actin system dynamics and MMPs has been speculated because it has been observed that cytoskeleton changes often precede MMPs modulation and actin microfilament dynamics might be linked to the expression of MMP genes [65]. Regarding collagen I, contrary to what observed for FN, in stimulated fibroblasts it showed an intracellular localization and no extracellular fibrils were observed. If an increase in MMP-1 expression following IL-1β and/or TNF-α stimulation has been unanimously reported [39,66,67,68], the effects the two cytokines have on collagen synthesis remain uncertain. Many studies reported that both IL-1β and TNF-α inhibit collagen I synthesis [67,69,70,71], but other studies demonstrated that IL-1β and TNF-α increased collagen I synthesis in human renal fibroblasts [72] and in murine intestinal myofibroblasts [38], respectively. A proposed scenario is that, in some conditions, the antifibrotic effect of TNF-α is overwhelmed by its central role in driving inflammation [66].

Fibroblasts stimulated with IL-1β and TNF-α and then exposed to NIR laser radiation recovered features more similar to unstimulated controls as regards the expression and distribution of α5β1, FN, collagen I, and MMP-1. Therefore, laser treatment was also able to counteract the cytokine effects on α5β1 integrin and the proteins involved in ECM turnover and remodeling after injury. It is noteworthy that, in the case of FN, not only the expression returned to levels comparable with unstimulated controls, but in samples treated with laser radiation the fibrils showed a more ordered and parallel distribution. This effect of laser radiation on FN and collagen fibril organization has already been described [73] and could be connected with the laser radiation’s ability to prevent fibrotic scars. The influence of red-NIR laser radiation on the expression of α5β1 integrin, FN, collagen, and MMP-1 has already been investigated in studies concerning laser application in the management of inflammatory response and wound healing. The results of these studies are controversial. A recent study on a model of diabetic wounded fibroblast cells showed that PBMT (660 nm wavelength) downregulated the expression of the genes *FN1*, *ITGA5*, and *ITGB1*, encoding for FN, α5, and β1 integrin subunits, respectively [74]. In a study on an immunosuppressed rat wounded model, PBMT by an 810 nm pulsed laser induced an increase in FN expression [75]. Enhanced FN expression was found also in human fibroblasts irradiated with a 940 nm diode laser [58]. A research on the effects of different protocols of PBMT in the healing of open wounds in rats showed that all the protocols used induced an increase in collagen deposition, but at different extent, depending on wavelength and fluence applied [43]. Using similar fluence but a different wavelength, a decrease in collagen production was found in wounded human skin fibroblasts [76]. In a rat model of wound healing, collagen deposition did not increase 3 days after laser treatment, but it increased significantly at day 7 after treatment [77]. Sakata et al. [28] found that, in chondrocytes stimulated with IL-1β, MMP-1 increased and then decreased after NIR irradiation applied post-stimulation, in complete agreement with what has been observed in the present study on fibroblasts activated by IL-1β and TNF-α.

From the outcomes of the studies mentioned above, it is evident that expression and function of α5β1 integrin, FN, collagen, and MMP-1 can be modulated through application of PBMT. However, results so uneven as those reported in the literature about PBMT effects demonstrate once again that it is very difficult to compare studies carried out using different experimental models, laser sources, and treatment parameters. Laser source and treatment protocol should be characterized for their biological effects before application for the management of specific pathological conditions.

In this paper, an in vitro model of fibroblast activation via stimulation with the pro-inflammatory cytokines IL-1β and TNF-α has been proposed and used to test the anti-inflammatory effect of a dual wavelength NIR laser source widely used in clinics to promote healing and reduce inflammation and pain. Like all in vitro models, a limit of the proposed model is to provide a very partial representation of what happens in vivo during inflammation and healing (a single cell population and two pro-inflammatory cytokines vs. many cell populations and a plethora of pro- and anti-inflammatory molecules). However, it can be considered representative of the early stage of the inflammation phase after an injury, when M1 macrophages produce great amounts of IL-1β and TNF-α. In the normal evolution of inflammation, macrophage phenotype is expected to shift from M1 to M2, with increased TGF-β production and a decrease in IL-1β and TNF-α levels [60]. Therefore, the proposed model can be considered also representative of altered evolution of inflammation with persistence of high levels of TNF-α, compared to TGF-β levels, due to the failure to switch from M1 to M2. 

Using this model, the effectiveness of PBMT by a dual wavelength NIR laser source (MLS-MiS) in reducing inflammation has been tested, and the results obtained show that PBMT, administered through the laser source and protocol here described, is significantly effective in preventing the effects of IL-1β and TNF-α, thus modulating the cell inflammatory response and favoring cell return to the basal physiological state. 

The anti-inflammatory effect of red-NIR laser radiation has been already reported in a number of studies but, to the best of our knowledge, it is the first time that it is evaluated and confirmed in an “in vitro” model of IL-1β and TNF-α activated dermal fibroblasts. Moreover, the significant anti-inflammatory activity of the laser emission tested in the present research is consistent with our previous studies carried out with the same laser source. In an in vitro model of myoblasts, it was found to increase the expression of NLRP10, a potent inhibitor of inflammasome activation and IL-1β and IL-18 production, as well as that of PP1, which regulates many important cell functions and favors cell recovery from stress to basal state [25]. In vivo, the same laser emission was able to reduce inflammatory infiltrate and accelerate the healing of ulcers in feline stomatitis [78] while, in a rat model of neuropathic pain induced by trauma, it significantly lowered inflammation and pain and preserved the myelin sheath [79]. It is well known that, when released by cells under pro-inflammatory stimuli, IL-1β and IL-18 induce the production of other pro-inflammatory cytokines, such as interferon-γ (INF γ), TNFα, IL-6, etc., thus triggering a cascade of events which further increase and perpetuate inflammation. Therefore, the ability of the proposed laser treatment to inhibit IL-1β and IL-18 release, through increased NLRP10 production, could explain its effectiveness in controlling fibroblast activation induced by IL-1β and TNF-α stimulation, thus damping excessive inflammatory response. Further studies could help to define treatment protocols specific for each different healing phase. 

## Figures and Tables

**Figure 1 biomedicines-09-00307-f001:**
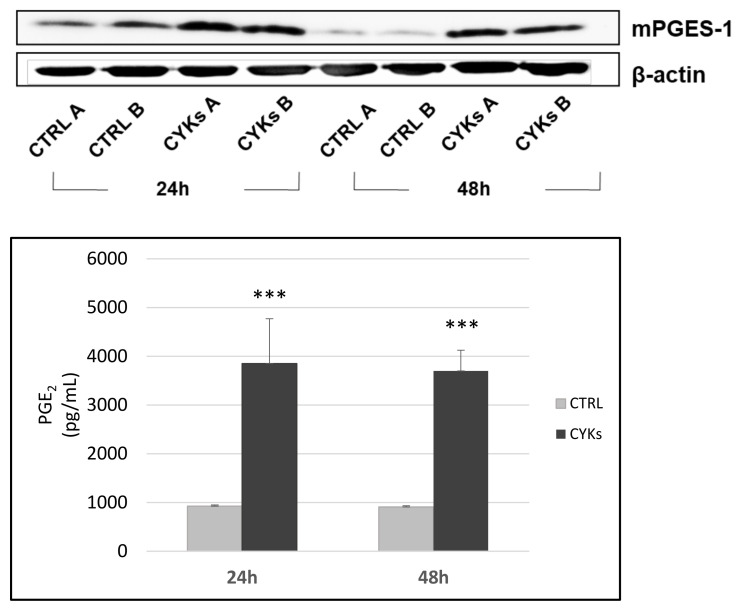
Development of an inflammation model on normal human dermal fibroblast (NHDF) cells. Fibroblasts were treated with IL-1β (10 ng/mL) + TNF-α (10 ng/mL) for 24 h and 48 h. Whole cell lysates were collected to assess mPGES-1 expression by Western blot (upper panel). Samples A and B represent intra-experimental duplicates. The measurement of PGE-2 performed by immunoenzymatic assay is reported (lower panel). At both times, there is an upregulation of the prostanoid system. Data represent means +/− SD (*n* = 3) *** *p* < 0.001 CYKs group vs. CTRL group.

**Figure 2 biomedicines-09-00307-f002:**
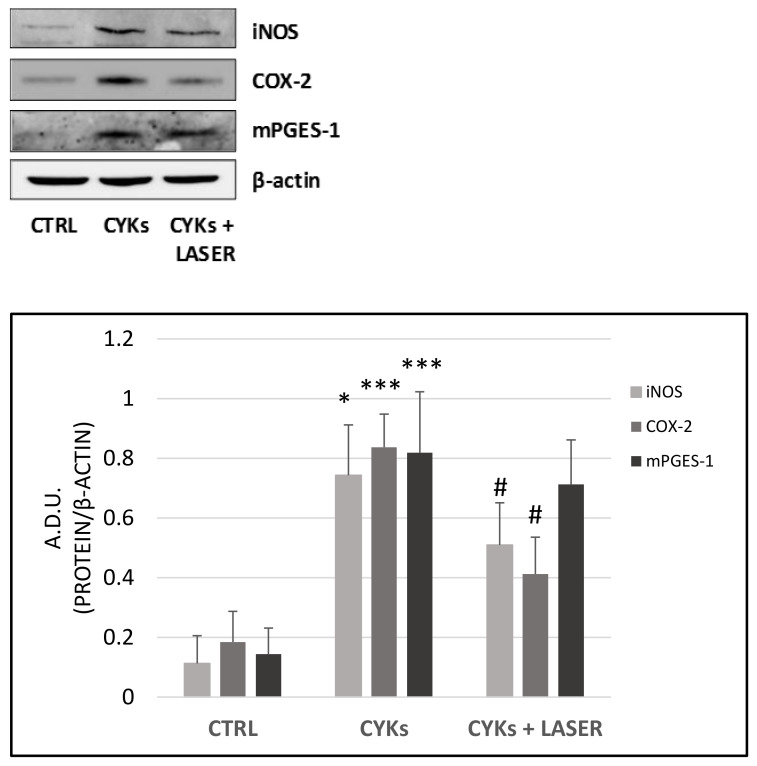
Effect of laser treatment on NDHF cells stimulated with pro-inflammatory cytokines. Fibroblasts were treated with IL-1β and TNF-α, each at 10 ng/mL for 48 h, then culture medium was replaced by fresh culture medium and samples divided into 3 groups: CYKs + Laser—samples stimulated with cytokine mix and then exposed to laser treatments (3 treatments, repeated once a day, for 3 consecutive days); CYKs—samples stimulated with cytokine mix and not exposed to laser treatment; CTRL—samples not stimulated and not exposed to laser treatment. Six hours after the third laser treatment, whole cell lysates of all samples were collected, and Western blot was performed to assess protein abundance of iNOS, COX-2, and mPGES-1 (upper panel). Immunoblots were analyzed by densitometry and the results, expressed as arbitrary density units (A.D.U.), were normalized to β-Actin (lower panel). Data represent means +/− SD (*n* = 2) * *p* < 0.05 and *** *p* < 0.001 CYKs group vs. CTRL group, # *p* <0.05 CYKs + Laser group vs. CYKs group.

**Figure 3 biomedicines-09-00307-f003:**
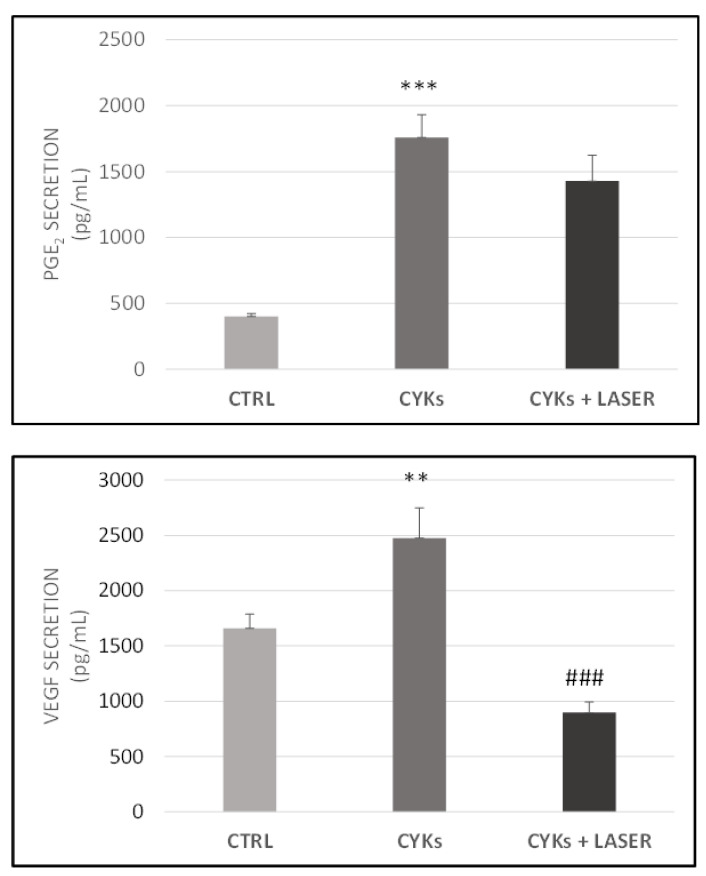
Modulation of PGE-2 and vascular endothelial growth factor (VEGF) release in NHDF cells exposed to pro-inflammatory cytokines and laser treatment. Fibroblasts were treated with IL-1β and TNF-α, each at 10 ng/mL for 48 h, then the culture medium was replaced by fresh culture medium and samples divided into 3 groups: CYKs + Laser—samples stimulated with cytokine mix and then exposed to laser treatments (3 treatments, repeated once a day, for 3 consecutive days); CYKs—samples stimulated with cytokine mix and not exposed to laser treatment; CTRL—samples not stimulated and not exposed to laser treatment. Six hours after the third laser treatment, all samples were recovered, and PGE-2 (upper panel) and VEGF (lower panel) levels were evaluated in conditioned media using specific ELISA kits. Dosing of each condition was performed in double, and quantification is expressed as pg/mL. Data represent means +/− SD (*n* = 2) ** *p* < 0.01 and *** *p* < 0.001 CYKs group vs. CTRL group, ### *p* < 0.001 CYKs + Laser group vs. CYKs group.

**Figure 4 biomedicines-09-00307-f004:**
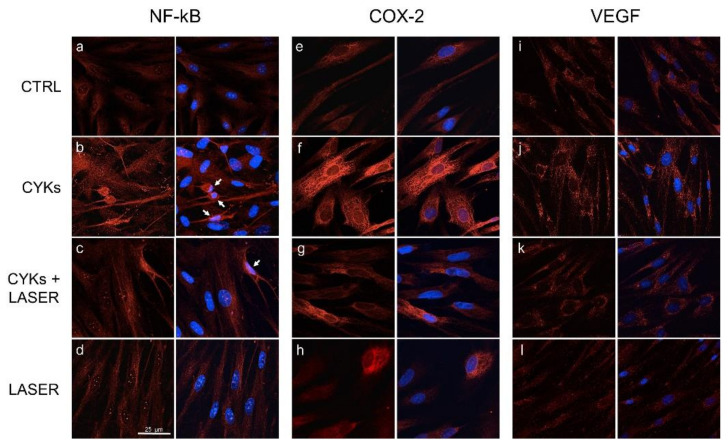
Laser treatment reduces inflammatory response in NHDF cells by limiting NF-κB translocation into the nucleus and down-regulating COX-2 and VEGF expression. Confocal analysis of NF-κB (panels (**a**–**d**)), COX-2 (panels (**e**–**h**)) and VEGF (panels (**i**–**l**)) expression and localization (magnification 63×) evaluated by immunofluorescence on NHDF in basal conditions (CTRL; panels (**a**,**e**,**i**)), stimulated with IL-1β and TNF-α for 48 h (CYKs; panels (**b**,**f**,**j**)), stimulated with IL-1β and TNF-α for 48 h, and then exposed to laser treatments (3 treatments, repeated once a day, for 3 consecutive days) (CYKs + LASER; panels (**c**,**g**,**k**)) and exposed to laser alone (LASER; panels (**d**,**h**,**l**)). For each series, the left panels show the protein of interest in red, while DAPI staining (blue) was merged on the right panels. White arrows indicate cells with nuclear localization of NF-kB. Bar = 25 µm.

**Figure 5 biomedicines-09-00307-f005:**
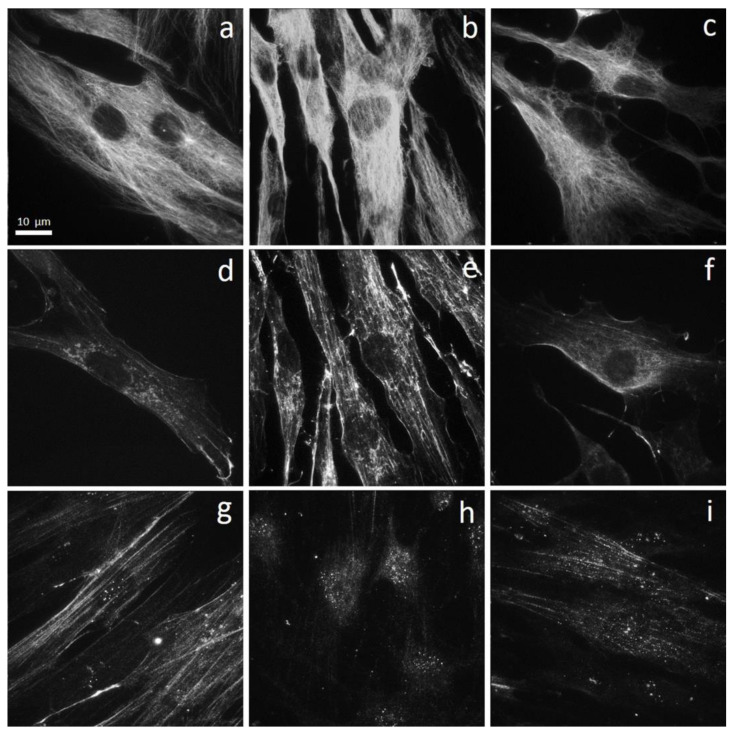
Effect of laser treatment on tubulin, α-actin, α-SMA expression, and distribution. Microscopy analysis of tubulin (**a**-**c**), α-actin (**d**–**f**), and α-SMA (**g**–**i**) expression evaluated by immunofluorescence (magnification 100×) on NHDF in basal conditions (CTRL; panels (**a**,**d**,**g**)), stimulated with IL-1β and TNF-α for 48 h (CYKs; panels (**b**,**e**,**h**)), stimulated with IL-1β and TNF-α for 48 h, and then exposed to laser treatments (3 treatments, repeated once a day, for 3 consecutive days) (CYKs + Laser; panels (**c**,**f**,**i**)). Bar = 10 µm.

**Figure 6 biomedicines-09-00307-f006:**
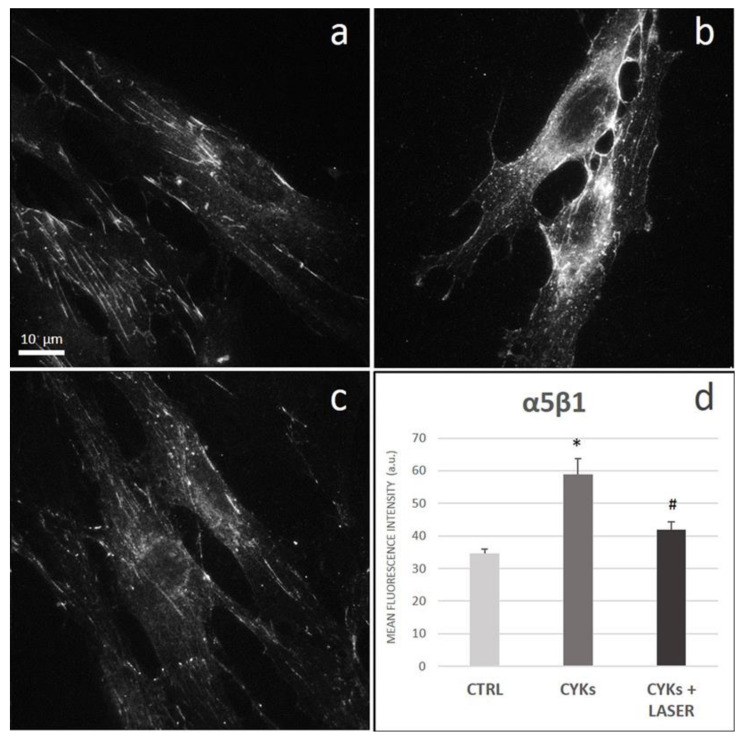
Effect of laser treatment on α5β1 expression and distribution. Microscopy analysis of α5β1 expression evaluated by immunofluorescence (magnification 100×) on NHDF in basal conditions (CTRL; panel (**a**)), stimulated with IL-1β and TNF-α for 48 h (CYKs; panel (**b**)), stimulated with IL-1β and TNF-α for 48 h and then exposed to laser treatments (3 treatments, repeated once a day, for 3 consecutive days) (CYKs + Laser; panel (**c**)). Bar = 10 µm. The histogram reports the mean pixel intensity, acquired by ImageJ software after appropriate thresholding and subsequent image masking (panel (**d**)). * *p* < 0.05 CYKs group vs. CTRL group; # *p* < 0.05 CYKs + Laser group vs. CYKs group (*n* = 3).

**Figure 7 biomedicines-09-00307-f007:**
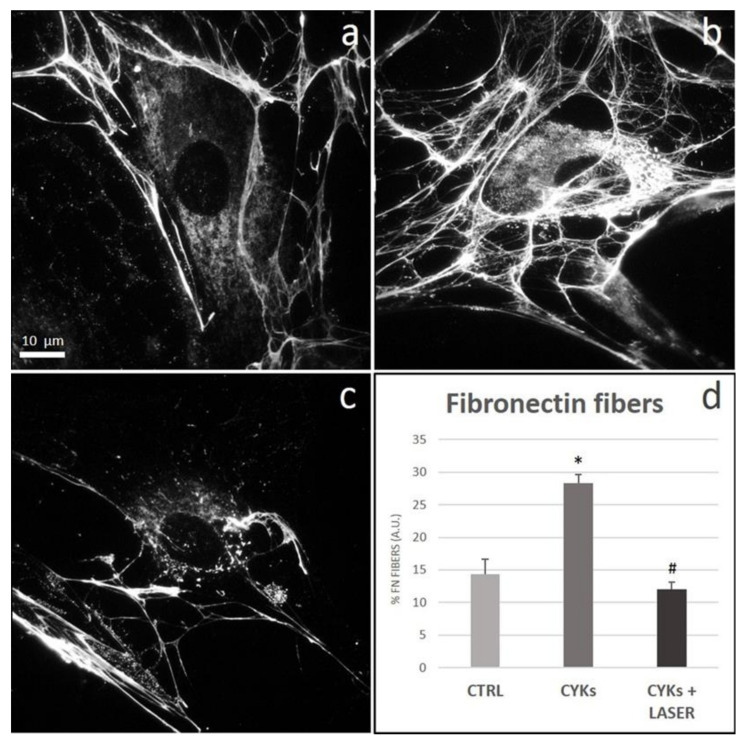
Effect of laser treatment on fibronectin expression and organization. Microscopy analysis of fibronectin expression evaluated by immunofluorescence (magnification 100×) on NHDF in basal conditions (CTRL; panel (**a**)), stimulated with IL-1β and TNF-α for 48 h (CYKs; panel (**b**)), stimulated with IL-1β and TNF-α for 48 h and then exposed to laser treatments (3 treatments, repeated once a day, for 3 consecutive days) (CYKs + Laser; panel (**c**)). Bar = 10 µm. The histogram reports the % of the surface area with fibers, acquired by ImageJ software after appropriate thresholding to only include the stained fibers (panel (**d**)). * *p* < 0.05 CYKs group vs. CTRL group; # *p* < 0.05 CYKs + Laser group vs. CYKs group (*n* = 3).

**Figure 8 biomedicines-09-00307-f008:**
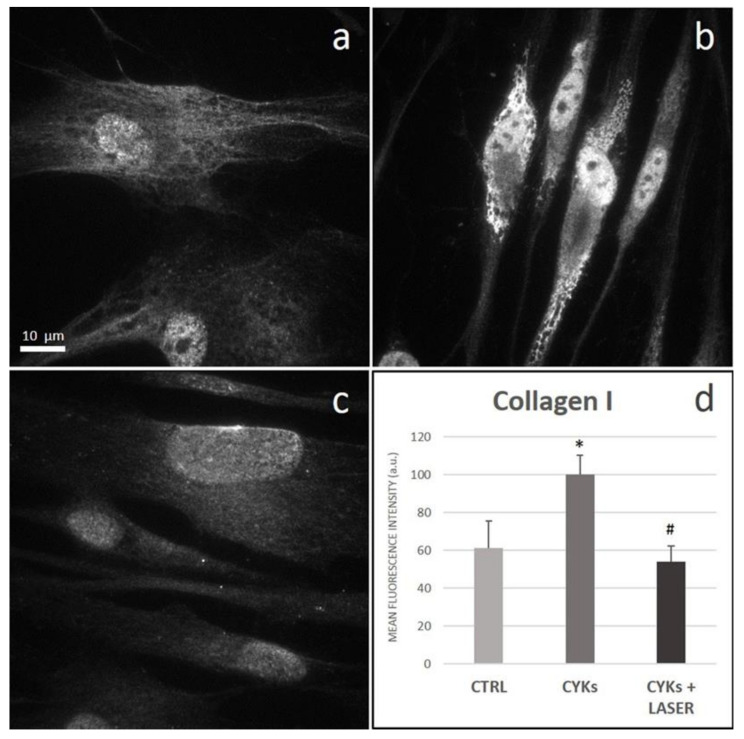
Effect of laser treatment on Collagen I expression and distribution. Microscopy analysis of Collagen I expression evaluated by immunofluorescence (magnification 100×) on NHDF in basal conditions (CTRL; panel (**a**)), stimulated with IL-1β and TNF-α for 48 h (CYKs; panel (**b**)), stimulated with IL-1β and TNF-α for 48 h and then exposed to laser treatments (3 treatments, repeated once a day, for 3 consecutive days) (CYKs + Laser; panel (**c**)). Bar = 10 µm. The histogram reports the mean pixel intensity, acquired by ImageJ software after appropriate thresholding and subsequent image masking (panel (**d**)). * *p* < 0.05 CYKs group vs. CTRL group; # *p* < 0.05 CYKs + Laser group vs. CYKs group (*n* = 3).

**Figure 9 biomedicines-09-00307-f009:**
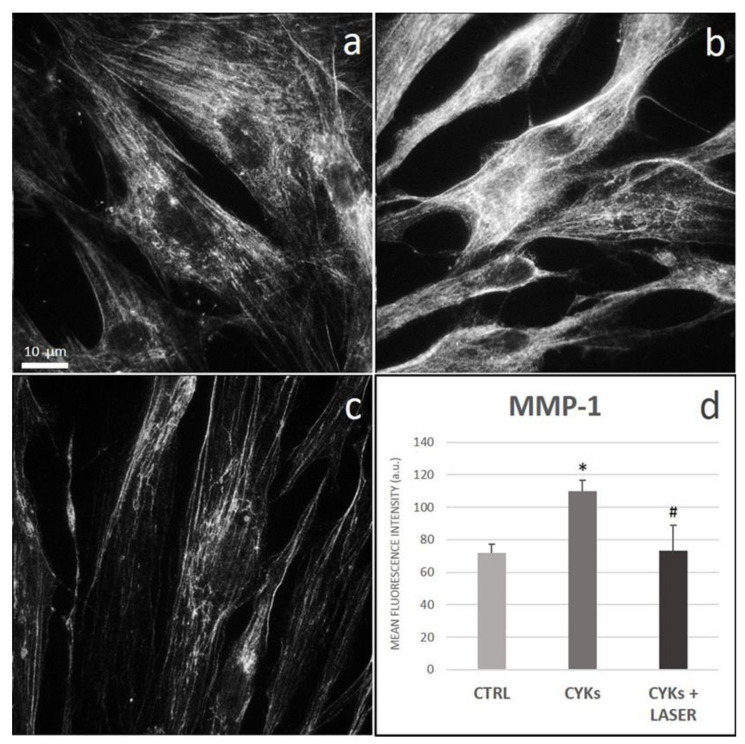
Effect of laser treatment on MMP-1 expression and distribution. Microscopy analysis of MMP-1 expression evaluated by immunofluorescence (magnification 100×) on NHDF in basal conditions (CTRL; panel (**a**)), stimulated with IL-1β and TNF-α for 48 h (CYKs; panel (**b**)), stimulated with IL-1β and TNF-α for 48 h and then exposed to laser treatments (3 treatments, repeated once a day, for 3 consecutive days) (CYKs + Laser; panel (**c**)). Bar = 10 µm. The histogram reports the mean pixel intensity, acquired by ImageJ software after appropriate thresholding and subsequent image masking (panel (**d**)). * *p* < 0.05 CYKs group vs. CTRL group; # *p* < 0.05 CYKs + Laser group vs. CYKs group (*n* = 3).

## Data Availability

The data sets generated and/or analyzed during the current study are available from the corresponding author on reasonable request.

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
