# Peer review of "Effect of NIR Laser Therapy by MLS-MiS Source on Fibroblast Activation by Inflammatory Cytokines in Relation to Wound Healing"

_biomedicines, 2021, doi:10.3390/biomedicines9030307_

Round 1

Reviewer 1 Report

The manuscript entitled “Effect of NIR laser therapy by MLS-MiS source on fibroblast 2 activation by inflammatory cytokines in relation to wound healing” provides new insight on NIR laser therapy to control fibroblast activation. The novelty in this manuscript is well written with detailed description and sheds new light on the NIR laser therapy. Certainly, this study is interesting and significant, however, there are some points which the author’s need to be addressed.

Comments

In Fig. 2 The cytokines +  laser group showed a strong decrease in inflammatory enzyme expression compared to the group stimulated only with cytokines, but the laser group is missing. Please include this group.

It has been shown by other groups that laser treatment activates the NF-kB pathway in fibroblast cells. Here, the authors showed that Laser treatment reduces the inflammatory response in NHDF cells by limiting NF-κB translocation into the nucleus and down-regulating COX-2 and VEGF expression but this NF-kB and VEGF data are not convincing. The authors should include the laser treatment group and show at the protein level.

Is laser exposure decrease PEG2 and VEGF secretion to a similar extent as a combined group?

Does Laser treatment show any effect on the Microsomal PGE synthase-1 (mPGES-1) level?

It would be good to include laser group in other experiments if possible.

Author Response

The manuscript entitled “Effect of NIR laser therapy by MLS-MiS source on fibroblast activation by inflammatory cytokines in relation to wound healing” provides new insight on NIR laser therapy to control fibroblast activation. The novelty in this manuscript is well written with detailed description and sheds new light on the NIR laser therapy. Certainly, this study is interesting and significant, however, there are some points which the author’s need to be addressed.

Answer: we thank the reviewer for the positive comments and inputs.

Comments

In Fig. 2 The cytokines +  laser group showed a strong decrease in inflammatory enzyme expression compared to the group stimulated only with cytokines, but the laser group is missing. Please include this group.

It has been shown by other groups that laser treatment activates the NF-kB pathway in fibroblast cells. Here, the authors showed that Laser treatment reduces the inflammatory response in NHDF cells by limiting NF-κB translocation into the nucleus and down-regulating COX-2 and VEGF expression but this NF-kB and VEGF data are not convincing. The authors should include the laser treatment group and show at the protein level.

Is laser exposure decrease PEG2 and VEGF secretion to a similar extent as a combined group?

Does Laser treatment show any effect on the Microsomal PGE synthase-1 (mPGES-1) level?

It would be good to include laser group in other experiments if possible.

Answer: The effect of laser treatment alone was performed in preliminary experiments performed during the initial setting of the experimental procedures and protocol. Results did not document any significant change in cell behavior and morphology. In the further experiments the condition of laser treatment alone was not performed except for the immunofluorescence detection. In the revised version the effect of laser on COX-2, NF-KB and VEGF is reported (see new Figure 4) documenting that laser alone induce some NF-kB translocation to the nucleus and that there was no change in COX-2 and VEGF labeling respect to control condition. The related materials and results have been added and discussed in the revised version of the manuscript.

Reviewer 2 Report

In this manuscript, the authors intended exploit photobiomodulation therapy as anti-inflammatory treatment. The authors used normal human dermal fibroblasts as model for its study and performed an exhaustive characterization of several inflammatory parameters. The manuscript is overall well written and well organized and the topic is of high interest. However, I found major and minor issues that should be addressed before publication.

Major topics:

  • What is the novelty of the research in comparison to the other studies described in the manuscript, some with more modern approaches, such as 3D cultures? While it can be understandable after reading the whole manuscript and the references, it should be more easily depicted.
  • I do miss a positive control for anti-inflammatory activity in all studies. It is of very importance that the behaviour observed should be present in many of the results from this positive control. It will give more strength to the work.
  • Why do mPGES-1 and PGE-2 do not reduce significantly as the other test subjects? That part is missing from the discussion.

Minor topics:

  • The introduction is too extensive. Some of the discussed topics should be simplified. Furthermore, the first part of the introduction is lacking references.
  • There are small incoherencies through the text that should be corrected (e.g., line 524 of page 15 is missing text; line 497-498 of page 14 has an understandable sentence)
  • Units should be verified to be coherent through the whole manuscript (e.g., the unit h is both with and without a space between the number and the unit)
  • Figure 3 – the figure caption should be corrected for the significance of the statistical analysis
  • There are too much figures in the manuscript. Figures 5,6 and 7, for instance, could be fused.
  • Figure 11 – The control (a) seems to have, visually, more fluorescence than the treatment (c) sample. However, in the fluorescence analysis (d) that is not the case. Could you explain?
  • There are some parts of the discussion that are simply a collection of the literature but with no further correlation to the results obtained. Please simplify.

Author Response

In this manuscript, the authors intended exploit photobiomodulation therapy as anti-inflammatory treatment. The authors used normal human dermal fibroblasts as model for its study and performed an exhaustive characterization of several inflammatory parameters. The manuscript is overall well written and well organized and the topic is of high interest. However, I found major and minor issues that should be addressed before publication.

Answer: the comments and careful reading of our manuscript are really appreciated.

Major topics:

  • What is the novelty of the research in comparison to the other studies described in the manuscript, some with more modern approaches, such as 3D cultures? While it can be understandable after reading the whole manuscript and the references, it should be more easily depicted.

Answer: the novelty of the paper has been stressed at the end of the introduction.

  • I do miss a positive control for anti-inflammatory activity in all studies. It is of very importance that the behaviour observed should be present in many of the results from this positive control. It will give more strength to the work.

Answer: while we understand the point raised by the referee, it was difficult to define the proper anti-inflammatory drug to be used as real control: antibodies against the single cytokines?  A NSAID able to block the downstream COX activity? A steroid anti-inflammatory drug which is upstream of the inflammatory cytokines? Considering these limits, we decided to use the basal unstimulated condition as clean control respect to the multi-factorial actions of laser therapy, as evidenced by our molecular/morphological findings. On top of this, it was not feasible to repeat most of the experiments in the week given for revision.

  • Why do mPGES-1 and PGE-2 do not reduce significantly as the other test subjects? That part is missing from the discussion.

Answer: the discussion of these data has been added (see page 19 lines 11-13).

Minor topics:

  • The introduction is too extensive. Some of the discussed topics should be simplified. Furthermore, the first part of the introduction is lacking references.

Answer: the introduction has been accordingly revised by shortening the first paragraphs. Since the article is part of the special issue on wound healing, we prefer to give some more detail on the molecules and pathways involved.

  • There are small incoherencies through the text that should be corrected (e.g., line 524 of page 15 is missing text; line 497-498 of page 14 has an understandable sentence)

Answer: we thank the referee for the suggestions and the sentences have been corrected for punctuation and meaning.

  • Units should be verified to be coherent through the whole manuscript (e.g., the unit h is both with and without a space between the number and the unit)

Answer: the whole manuscript has been checked for incoherencies.

  • Figure 3 – the figure caption should be corrected for the significance of the statistical analysis

Answer: we thank the referee for the suggestion: the legend has been corrected.

  • There are too much figures in the manuscript. Figures 5,6 and 7, for instance, could be fused.

Answer: As suggested, we condensed figure 5, 6 and 7 in the new figure 5. The legends and text have been corrected accordingly.

  • Figure 11 – The control (a) seems to have, visually, more fluorescence than the treatment (c) sample. However, in the fluorescence analysis (d) that is not the case. Could you explain?

Answer: The higher fluorescence intensity was found in panel b) respect to a) and c). Please consider that  while images are representative, the quantification of fluorescence has been done on 30 cells coming from 10 different random fields/slide. At least 3 independent experiments in triplicate were run for each parameter considered.

  • There are some parts of the discussion that are simply a collection of the literature but with no further correlation to the results obtained. Please simplify.

Answer: the discussion has been revised as suggested and correlation between literature cited and results obtained has been highlighted.

Round 2

Reviewer 1 Report

The author should respond to comments point by point and address all the raised comments.

Reviewer 2 Report

The authors have improved the manuscript to an acceptable form, with just some minor formatting typos that I believe it is just regarding the corrected version I received.